# Assessment of malaria infection among pregnant women and children below five years of age attending rural health facilities of Kenya: A cross-sectional survey in two counties of Kenya

**Collins Okoyo**[1]*, **Edward Githinji**[1], **Ruth W. Muia**[2], **Janet Masaku**[1], **Judy Mwai**[3], **Lilian Nyandieka**[3], **Stephen Munga**[4], **Sammy M. Njenga**[1], **Henry M. Kanyi**[1]

1 Eastern and Southern Africa Centre of International Parasite Control, Kenya Medical Research Institute (KEMRI), Nairobi, Kenya, 2 Division of Family Health, Ministry of Health (MOH), Nairobi, Kenya, 3 Centre for Public Health Research, Kenya Medical Research Institute (KEMRI), Nairobi, Kenya, 4 Centre for Global Health Research, Kenya Medical Research Institute (KEMRI), Nairobi, Kenya

* comondi@kemri.org, collinsomondiokoyo@gmail.com

**Data Availability Statement:** All relevant data supporting the conclusions of this paper are

## Abstract

### Background

In Kenya, health service delivery and access to health care remains a challenge for vulnerable populations, particularly pregnant women and children below five years. The aim of this study, therefore, was to determine the positivity rate of *Plasmodium falciparum* parasites in pregnant women and children below five years of age seeking healthcare services at the rural health facilities of Kwale and Siaya counties as well as their access and uptake of malaria control integrated services, like antenatal care (ANC), offered in those facilities.

### Methods

Cluster random sampling method was used to select pregnant women and children below five years receiving maternal and child health services using two cross-sectional surveys conducted in eleven rural health facilities in two malaria endemic counties in western and coastal regions of Kenya. Each consenting participant provided single blood sample for determining malaria parasitaemia using microscopy and polymerase chain reaction (PCR) techniques.

### Results

Using PCR technique, the overall malaria positivity rate was 27.9% (95%CI: 20.9–37.2), and was 34.1% (95%CI: 27.1–42.9) and 22.0% (95%CI: 13.3–36.3) in children below five years and pregnant women respectively. Additionally, using microscopy, the overall positivity rate was 39.0% (95%CI: 29.5–51.6), and was 50.4% (95%CI: 39.4–64.5) and 30.6% (95%CI: 22.4–41.7) in children below five years and pregnant women respectively. Siaya County in western Kenya showed higher malaria positivity rates for both children (36.4%

provided within the article. The raw datasets analyzed are available from ESACIPAC, KEMRI (www.kemri.org) upon request.

**Funding:** The financial support for this research was provided by the Government of Kenya through KEMRI grant number LO656 for the field-based work and grant number LO660 for the PCR work. The funders had no role in the study design, data collection and analysis, decision to publish or preparation of the manuscript.

**Competing interests:** The authors have declared that no competing interests exist.

**Abbreviations:** ANC, antenatal care; aOR, adjusted odds ratio; CBRD, Centre for Biotechnology Research and Development; CIs, confidence interval; DBS, dried blood spot; DNA, deoxyribonucleic acid; HIV/AIDS, human immunodeficiency virus/acquired immunodeficiency syndrome; IPTp-SP, intermittent preventive treatment with sulfadoxine-pyrimethamine; KEMRI, Kenya Medical Research Institute; LLINs, long-lasting insecticidal nets; NMCP, national malaria control programmes; NMS, national malaria strategy; ODK, open data kit; OR, odds ratio; PCR, polymerase chain reaction; SD, standard deviation; WHO, World Health Organization.

and 54.9%) and pregnant women (27.8% and 38.5%) using both PCR and microscopy diagnosis techniques respectively, compared to Kwale County that showed positivity rates of 27.2% and 37.9% for children and 5.2% and 8.6% for pregnant women similarly using both PCR and microscopy techniques respectively. Pregnant women presenting themselves for their first ANC visit were up to five times at risk of malaria infection, (adjusted odds ratio = 5.40, 95%CI: 0.96–30.50, p = 0.046).

## Conclusion

Despite evidence of ANC attendance and administration of intermittent preventive treatment with sulfadoxine-pyrimethamine (IPTp-SP) dosage during these visits, malaria positivity rate was still high among pregnant women and children below five years in these two rural counties. These findings are important to the Kenyan National Malaria Control Programme and will help contribute to improvement of policies on integration of malaria control approaches in rural health facilities.

## Introduction

Currently, the World Health Organization (WHO) estimates that 3.2 billion people are at risk of malaria infection globally. Based on the World Malaria Report of 2017 [1], there were 216 million cases of malaria infection in 2016 up from 211 million cases in 2015 with estimated deaths of 445,000 in 2016. The Sub-Saharan Africa region contributes to disproportionately high share of the global malaria burden with the region harboring approximately 90% of all malaria infection cases and 91% of all malaria deaths in 2016 [1].

Malaria infection is disproportionately higher in some groups of the population such as infants, children under the age of five years, pregnant women and patients with underlying diseases like human immunodeficiency virus, acquired immunodeficiency syndrome (HIV/AIDS), as well as non-immune migrants, mobile populations and travelers [1]. The WHO has encouraged National Malaria Control Programmes (NMCP) of the heavily burdened countries to put in place special measures to protect these vulnerable groups of the population from malaria infection, noting their specific circumstances [2,3].

In Kenya, malaria remains a major public health problem and cause of high morbidity and mortality with over 70% of her population being at risk of the infection [4]. The burden of the infection in Kenya is non-homogenous with areas surrounding Lake Victoria and the Coastal region representing the highest risk areas with children below five years of age and pregnant women being the most vulnerable to the infection [5,6]. Previous studies conducted in Kenya have indicated varying levels of malaria positivity rates, including hospitalized malaria cases, and which generally range between 4.8% to 70% among children below five years [7,8], and between 1% to 60% among pregnant women [9–11]. However, there is still insufficient surveillance data on malaria infection burden and prevalence among these two vulnerable populations, and this study results contribute to this urgent need of surveillance data. Surveillance data are usually vital in providing control programmes with the ability to track the number of malaria cases, identify upsurges and monitor the spread of infection so that they could offer interventions effectively [12].

Malaria infection in pregnancy is a significant public health problem in Kenya and is associated with considerable morbidity and mortality for the pregnant women and their infants

[4,13]. The associated symptoms and complications of malaria during pregnancy fluctuates based on a given geographical location, level of acquired immunity of an individual and transmission intensity in a particular area. In areas of high-transmission, where levels of acquired immunity to *Plasmodium falciparum* infection tend to be high, malaria is generally asymptomatic [14]. In such areas, parasites may be present in the placenta during pregnancy and sometimes lead to maternal anaemia even in the absence of documented peripheral parasitaemia [14]. Both anaemia and placental parasitaemia can contribute to low birth weight which is significantly associated with infant mortality [15]. Usually, in high-transmission areas, adverse effects of malaria infection during pregnancy are often common especially during the first pregnancy [2].

In low malaria transmission areas, where women of reproductive age with low acquired immunity, malaria during pregnancy is usually associated with anaemia, which increase the risk of severe malaria that may even lead to spontaneous abortion, stillbirth, premature birth and low birth weight. Hence, in such transmission areas, all pregnant women are highly vulnerable to severe malaria infection [16].

The Kenyan National Malaria Strategy (NMS) of 2009–2017, recommends the following package of interventions for the prevention and treatment of malaria infection during pregnancy; 1) use of long-lasting insecticidal nets (LLINs), 2) in all areas of moderate to high malaria transmission, provision of intermittent preventive treatment for malaria in pregnancy with sulfadoxine-pyrimethamine (IPTp-SP) during antenatal care (ANC) visits and 3) prompt diagnosis and effective treatment of malaria infections [4]. Interventions also include building the capacity of healthcare providers and strengthening the supply chain to deliver diagnostic tests and quality-assured medicines at all levels of the health systems.

Additionally, malaria in children under the age of five years is equally considered to be of public health concern and was estimated to account for approximately 70% of all malaria cases globally and 80% of all cases in Kenya in 2015 [4,17]. According to WHO, the number of deaths due to malaria in children under five years of age has significantly decreased since the year 2000 and thus malaria is no longer considered the leading cause of death in children in this particular age group, though, it still remains a major cause of morbidity in children in Sub-Saharan Africa region, where it accounts for 10% of all deaths of children [18].

In Kenya, provision and utilization of ANC services is common with over 94% of pregnant women reported to have received the services from skilled personnel in various levels of health facilities [19]. However, the spread of these services to the rural population and how they contribute to the reduction of malaria burden has not been fully explored. The aim of this study, therefore, was to determine the positivity rate of *Plasmodium falciparum* parasites in pregnant women and children below five years of age seeking healthcare services at the rural health facilities of Kwale and Siaya counties as well as their access and uptake of malaria control integrated services offered in those facilities.

## Materials and methods

### Study design and sampling

We conducted two cross-sectional descriptive surveys in Kwale and Siaya counties, lying within the moderate and high malaria endemic regions of coastal and western regions of Kenya, respectively. One rural health facility i.e. a dispensary, health centre or mobile clinic was randomly sampled from each sub-county of each of the two counties. The study populations included the pregnant women and children below five years of age attending ANC at the selected health facility. A cluster random sampling method was used to select a total of 180 pregnant women and an equal number of children under five years (n = 360) from five health

facilities (clusters) in Kwale County. Similarly, in Siaya County, a cluster random sampling method was used to select a total of 380 pregnant women and an equal number for children below five years (n = 760) from six health facilities (clusters). In each county, clusters were defined as the participating health facilities and they were randomly selected from each sub-county, thus in Kwale County the study used five clusters and six clusters in Siaya County. The achieved sample size per county was therefore allocated equally among the clusters. The sample size calculation was based on previously documented specific malaria prevalence, for each specific group, in the said regions [4], and calculated separately for each group of the participants in each county using the Fisher's fomular [20], stated as $n = \frac{Z^2 pq}{e^2}$; where Z is the score for a 5% type 1 error for a normal distribution (Z = 1.96), p is the prevalence of malaria in each different endemic region; taken as 13.5% in coast endemic region (Kwale County) [4], and 38.0% in lake endemic region (Siaya County) [4], conversely, q = 1-p, and e is the margin of error assumed as 5%. Further, from each group's minimum sample size calculated, we adjusted the final sample size by assuming a 10% non-response rate. Therefore, equal number of participants (either children below five years or pregnant women) were targeted in each cluster (health facility) within the respective county. The inclusion criteria for pregnant women were as follows; (1) a pregnant woman attending a selected health facility on the day of the survey and willing to provide a blood sample for malaria testing. Inclusion criteria for children were as follows; (1) a child attending the selected health facility on the day of the survey and who is below five years of age, (2) a child who is accompanied by a mother/caregiver and whose mother/caregiver provide consent to participate in the study, and (3) a child whose mother/caregiver provide consent for blood sample to be drawn from the child for malaria testing. Importantly, we note that the children, besides merely accompanying their mothers/caregivers to the health facility, some were brought to seek different services including immunization, post-natal checkups, and different forms of treatment, among other reasons.

## Data collection and survey procedures

The surveys were conducted between 15th and 24th May, 2017 in Kwale County, and between 28th June and 14th July, 2017 in Siaya County. These data collection periods coincided with the high malaria transmission season in Kenya which is usually estimated to be between April and July when also long rains are experienced [21,22]. After consenting, each participant (pregnant woman or child below five years of age) was asked to provide a finger prick blood sample for malaria testing using microscopy. This was conducted at the rural health facility using local facility-based laboratory technicians and supported by experienced laboratory technologists from the Kenya Medical Research Institute (KEMRI). Further, 60μl of blood was collected onto a filter paper for preparation of dried blood spots (DBS: 6 spots of 10μl each; TropBio Pty Ltd, Queensland, Australia). The DBS samples were transported to KEMRI laboratories in Nairobi for further molecular and serological examinations. Additionally, trained interviewers administered a pre-tested questionnaire to each participating child and pregnant woman regarding details related to their age, gender (for the case of children), and pregnancy and ANC visit (for the case of pregnant women). The questionnaire was partly adopted from a validated questionnaire routinely administered by the Malaria Indicator Survey (MIS) [4]. A copy of the study questionnaire, in English and translated versions of Kiswahili and Dholuo, has been included in the supplementary information (SI). All participants found positive for malaria using microscopy were referred for treatment at the health facility. All field staff were trained on the survey procedures and study protocol prior to the survey including mock surveys to pilot the study tools. It is important to note that both groups of participants (pregnant women or children below five years) were unpaired and were independent of each other, they

separately came to the health facility to seek services. All participants were enrolled in the study prior to them being attended to by the health facility staff.

## Sample processing and examination

**Sample processing and examination using microscopy.** Giemsa (3%) stained blood smears, both thick and thin were prepared and examined under light microscope. For the thick blood smear, a thick blood film was made on a dry glass slide and air dried on a staining rack for one hour. The thick blood smear was then dipped in 3% Giemsa stain, washed in distilled water for 5 minutes and left to dry. The thick blood smear was used to detect infection and estimate parasitaemia for they contain a greater amount of blood than the thin films and are therefore more likely to contain parasites. For the thin blood smear, a thin film of blood was made on a clean dry glass slide and left to dry. Two to three dips into absolute methanol ensured fixation of the thin smear which were left to air dry for 30 seconds. The slide was then flooded with 3% Giemsa stain for 30 minutes, rinsed with tap water and left to dry. The thin blood smear was later examined using microscope at 100x objectives in immersion oil to identify malaria species, quantify parasitaemia, and recognize parasites' asexual forms since the smear is thinly spread, fixed using absolute methanol hence better visualization of the parasites. Examination of these slides was conducted at the local health facility by trained local technicians and supported by experienced technologists from KEMRI, to identify and differentiate the malaria parasites. Parasitaemia was calculated as number of infected red blood cells per 1000 red blood cells counted in the thin film. A slide was declared negative if no malaria parasite was observed in 100 fields. Each microscopy slide was independently read as number of parasite counts in every two hundred white blood cells (counts/200 WBCs) by two malaria microscopy certified laboratory technicians. For purposes of quality control, 10% of all samples (both positive and negative) were randomly sampled and re-examined by a third malaria microscopy certified senior laboratory technologist.

**DNA extraction from the dried blood spots.** The template deoxyribonucleic acid (DNA) extraction from DBS was done using the DNeasy Blood & Tissue Kit (QIAGEN, Valencia, CA) according to the manufacturer's protocol. Purified DNA was eluted from the column with 50 microliter (μl) elution buffer and this DNA was stored at -20˚C before running nested polymerase chain reaction (PCR) assays. A negative control was included in every extraction cycle. A positive control pool of DNA was extracted independently from cultured *P. falciparum* 3D7 parasites at 1% parasitaemia obtained from the Malaria Laboratory in the Centre for Biotechnology Research and Development (CBRD), KEMRI. This positive control was used during all PCR assays alongside the test samples.

**Polymerase chain reaction (PCR).** PCR was performed according to Snounou *et al* [23] through a nested-protocol where a species-specific region of the 18s ribosomal DNA of *Plasmodium* was amplified. Primary amplification was conducted in a 25μl reaction mixture containing 5μl of genomic DNA, 1X DreamTaq Green PCR Master Mix (Thermo Scientific) having 2mM $MgCl_2$ concentration, 0.2mM dNTPs, 250nM of each primer and 0.625 U of Green Taq DNA polymerase. Reaction was performed in BIORAD MyCycler thermal cycler using a pair of primers targeting an outer region specific for the *Plasmodium* genus. The primers used for *Plasmodium* genus amplification were: rPLU5: 5'-CTT GTT GTT GCC TTA AAC TTC-3', and rPLU6: 5'-TTA AAA TTG TTG CAG TTA AAA CG-3'. One microlitre of the first PCR reaction product was used as a template in a secondary reaction targeting a region specific for *P. falciparum*. The temperature profile for the PCR was: five minutes at 95˚C; 25 cycles of one minute at 94˚C, two minutes at 55˚C, two minutes at 72˚C primary reaction; followed by the secondary reaction where the annealing temperature was modified to

58˚C and the cycles repeated 30 times. The primers used for *P. falciparum* detection were: rFAL1:5'-TTA AAC TGG TTT GGG AAA ACC AAA TAT ATT-3' and rFAL2: 5'-ACA CAA TGA ACT CAA TCA TGA CTA CCC GTC-3'. The PCR products were visualized under Ultra Violet light after 2% agarose gel electrophoresis in 1X Tris borate EDTA buffer and ethidium bromide staining. A sample was considered positive if a 205 base pairs band for *P. falciparum* was detected. In every set of reaction, negative and positive controls were included. A Fast-Ruler Low Range DNA Ladder (Thermo Scientific) was also used in gel electrophoresis. It is important to note that all samples collected in the field were re-examined using PCR technique and the laboratory staff performing PCR analysis were blinded to the microscopy results since they received samples labelled with identification numbers (IDs) only from the field staff.

## Ethics statement

The study protocol received ethical approval from KEMRI's Scientific and Ethics Review Unit (SERU No. 3252). Additional approval was provided by the county-level health authorities after they were appropriately briefed about the study. At health facility level, written participant consent for pregnant women and written parental consent for children below five years was obtained.

## Data management and analysis

Laboratory data reporting form and the questionnaires were programmed onto android-based smartphones and used to capture data electronically into the open data kit (ODK) system [24], which included in-built data quality checks to prevent data entry errors. The collected data were transmitted daily to the central server based in KEMRI Nairobi. *Plasmodium falciparum* infection was defined as a positive result, first by microscopy technique at the rural health facility and confirmed by PCR technique at the laboratory in Nairobi. As much as we have compared the malaria infection positivity rates by the two methods, overall emphasis was given to the PCR results since this technique has been confirmed to be highly sensitive, accurate and able to detect submicroscopic infections [25]. The overall malaria positivity rate, by the two methods, was calculated at the facility and county levels and the 95% confidence intervals (CIs) obtained using binomial regression model while adjusting for the study clusters. Overall, factors associated with malaria infection (detected using PCR technique) were first analyzed using univariable analysis and the strength of association measured as odds ratio (OR) using mixed effects logistic regression at 95%CIs and taking into account the study clusters. To select minimum adequate variables for multivariable analysis, an inclusion criterion of p-value <0.3 was pre-specified in a sequential (block-wise) variable selection method which selected covariates meeting the set criterion. Therefore, adjusted OR (aOR) were obtained by mutually adjusting all minimum generated variables using multivariable mixed effects logistic regression at 95%CIs and taking into account the study clusters. It is important to note that multivariable analysis was conducted among pregnant women only since there were few significant variables for children below five years to include in the multivariable model. This study presented the risk factor analysis using the PCR results only while noting that the same pattern of results were observed when fitted on microscopy results. Further, proportions were calculated for all other variables of interests, like the trimester of the first ANC attendance and IPTp-SP dosage uptake, at facility and county levels; and the statistical differences among the proportions by county was determined using test of proportions reporting the observed difference, Z-test statistic and p-value. All statistical analyses were carried out using STATA version 15.1 (STATA Corporation, College Station, TX, USA).

## Results

Overall, 1,128 respondents participated in the study, but 102 (9.0%) did not consent to malaria testing although they agreed to answer the survey exit questions and hence were excluded only from further analysis on malaria positivity rate. A total of 484 (47.2%) pregnant women and 542 (52.8%) children below five years participated in the study and provided a blood sample for malaria testing. DBS samples were collected for further molecular examination using PCR from 978 (86.7%) of the participants. Information on age was obtained from 1,013 (89.8%) of the participants and ranged from 14 to 43 years (SD = 6.3 years) among the pregnant women and 0 to 4 years (SD = 1.4 years) among the children below five years. With the exception of the pregnant women, gender information was collected for all the children below 5 years with male being 272 (50.5%) and female 267 (49.5%) (Table 1).

### Malaria positivity rates by microscopy and PCR techniques in the two study counties

Overall, the malaria positivity rate was 39.0% (95%CI: 29.5–51.6) by microscopy and 27.9% (95%CI: 20.9–37.2) by PCR. The positivity rate was higher in Siaya County 32.2% (95%CI: 24.0–43.1) than Kwale 16.5% (95%CI: 9.2–29.5) using PCR technique, and was the same case when using microscopy. In Siaya County, the highest overall positivity rate using PCR was observed in Ugenya Sub-County 51.9% (95%CI: 44.0–61.3) and least in Bondo Sub-County 19.8% (95%CI: 13.6–28.8). Similarly, in Kwale County, highest overall infection using PCR was in Matuga Sub-County 28.3% (95%CI: 18.9–42.4) and least in Lunga Lunga Sub-County 5.2% (95%CI: 1.7–15.6) (Table 1).

**Malaria in pregnant women.**   Using PCR technique, the overall malaria positivity rate among pregnant women was 22.0% (95%CI: 13.3–36.3). Siaya County accounted for almost twice the number of malaria infections as compared to Kwale County. According to sub-counties, Ugenya Sub-County recorded the highest number of positive cases at 54.7% (95%CI: 43.8–68.4) while Bondo Sub-County had the least number of cases at 12.8% (95%CI: 6.0–27.0), all in Siaya County. In Kwale County, Matuga Sub-County had the highest number of cases 9.1% (95%CI: 2.4–34.1), while Kinango Sub-County had least cases 3.3% (95%CI: 0.5–22.9). Most malaria positive women were aged between 38 to 43 years 39.1% (95%CI: 17.5–87.4) followed by those age between 14 to 19 years 24.2% (95%CI: 14.2–41.1) and least cases were among those aged 26 to 31 years 15.7% (95%CI: 7.4–33.4). By education level, pregnant women who had post-secondary level had no malaria positive cases but it was surprisingly highest among those with secondary level of education. Importantly, malaria positivity rate was highest among pregnant women who were in their first trimester 32.4% (95%CI: 18.3–57.5) and among those who were presenting themselves for their first ANC clinics 41.9% (95%CI: 24.5–71.6), as well as those who had not received any IPTp-SP dose 31.2% (95%CI: 17.1–56.9) (Table 1). The same pattern of infection was observed with microcopy results.

**Malaria in children below five years.**   Using PCR technique, the overall malaria positivity rate among children below five years was 34.1% (95%CI: 27.1–42.9), and the positivity rate increased with the age of the children. Similarly, Siaya County accounted for over half of the total malaria infections in children compared to Kwale County. Half of the infections in Siaya County were from Ugunja Sub-County 50.0% (95%CI: 39.3–63.6) with least of the infections in the county being from Bondo 25.0% (95%CI: 16.4–38.2) and Rarieda 25.0% (95%CI: 16.4–38.2). Similarly, almost half of the infections in Kwale County were from Kinango Sub-County 46.7% (95%CI: 31.8–68.4) with no cases observed in Lunga Lunga Sub-County. Overall, slightly more male children were positive for malaria 38.5% (95%CI: 28.4–52.0) compared to

**Table 1. Malaria positivity rate % (95%CI) in children below five years and pregnant women by microscopy and PCR techniques in Kwale and Siaya counties, Kenya.**

| Factor | Total number of facilities | Total number of participants (%) | Positive rate by microscopy % (95%CI) | | | Positive rate by PCR % (95%CI) | | | Positive rate by both microscopy and PCR % (95%CI) | | |
|---|---|---|---|---|---|---|---|---|---|---|---|
| | | | Overall (n = 1026) | Children below five years (n = 542) | Pregnant women (n = 484) | Overall (n = 978) | Children below five years (n = 528) | Pregnant women (n = 450) | Overall (n = 1013) | Children below five years (n = 540) | Pregnant women (n = 473) |
| **Overall** | **11** | **1128 (100%)** | **39.0 (29.5–51.6)** | **50.4 (39.4–64.5)** | **30.6 (22.4–41.7)** | **27.9 (20.9–37.2)** | **34.1 (27.1–42.9)** | **22.0 (13.3–36.3)** | **21.6 (15.5–30.1)** | **29.8 (23.7–37.6)** | **15.9 (9.8–25.5)** |
| *County* | | | | | | | | | | | |
| **Kwale County** | **5** | **362 (32.1%)** | **21.8 (13.7–34.7)** | **37.9 (22.9–62.7)** | **8.6 (5.6–13.2)** | **16.5 (9.2–29.5)** | **27.5 (14.5–52.5)** | **5.2 (3.2–8.5)** | **11.9 (4.7–30.2)** | **26.2 (13.3–51.8)** | **2.3 (1.1–5.1)** |
| Matuga | 1 | 61 (16.9%) | 41.0 (30.3–55.4) | 57.9 (44.1–75.9) | 13.6 (4.8–39.0) | 28.3 (18.9–42.4) | 39.5 (26.6–58.5) | 9.1 (2.4–34.1) | 26.2 (17.2–40.0) | 39.5 (26.6–58.5) | 4.5 (0.7–30.8) |
| Kinango | 1 | 60 (16.6%) | 26.7 (17.5–40.6) | 50.0 (35.0–71.5) | 3.3 (0.5–22.9) | 25.0 (16.1–38.8) | 46.7 (31.8–68.4) | 3.3 (0.5–22.9) | 25.0 (16.1–38.8) | 46.7 (31.8–68.4) | 3.3 (0.5–22.9) |
| Msambweni | 1 | 63 (17.4%) | 23.8 (15.3–37.0) | 43.3 (28.8–65.2) | 6.7 (1.7–25.4) | 15.0 (8.2–27.4) | 26.7 (14.7–48.3) | 3.6 (0.5–24.5) | 12.7 (6.6–24.3) | 26.7 (14.7–48.3) | 0 |
| Lunga lunga | 1 | 120 (33.2%) | 16.0 (10.6–24.1) | 12.9 (5.2–32.2) | 10.7 (3.7–31.2) | 5.2 (1.7–15.6) | 0 | 7.4 (2.0–28.1) | 0.9 (0.1–6.6) | 0 | 3.6 (0.5–24.5) |
| Mobile clinic* | 1 | 58 (16.0%) | 5.6 (1.8–16.7) | 6.3 (0.9–41.7) | 11.1 (3.0–41.0) | 2.9 (0.4–19.7) | 10.0 (1.6–64.2) | 0 | 1.9 (0.3–12.9) | 6.3 (0.9–41.7) | 0 |
| **Siaya County** | **6** | **766 (67.9%)** | **47.0 (37.1–59.6)** | **54.9 (41.5–72.7)** | **38.5 (31.6–46.9)** | **32.2 (24.0–43.1)** | **36.4 (28.5–46.5)** | **27.8 (17.3–44.4)** | **26.1 (19.9–34.2)** | **31.1 (24.6–39.5)** | **20.9 (14.0–31.2)** |
| Rarieda | 1 | 112 (14.6%) | 29.5 (22.1–39.2) | 32.8 (23.1–46.6) | 25.5 (15.7–41.6) | 22.0 (15.5–31.3) | 25.0 (16.4–38.2) | 18.2 (9.7–34.0) | 21.6 (15.2–30.8) | 25.0 (16.4–38.2) | 17.4 (9.3–32.6) |
| Gem | 1 | 132 (17.2%) | 38.6 (31.1–47.9) | 44.6 (34.0–58.5) | 32.8 (23.1–46.6) | 26.6 (19.9–35.4) | 33.8 (24.1–47.5) | 19.4 (11.6–32.2) | 19.8 (14.1–28.0) | 24.6 (16.1–37.7) | 15.9 (9.0–28.0) |
| Ugenya | 1 | 131 (17.1%) | 66.4 (58.8–75.0) | 86.2 (78.2–95.0) | 47.0 (36.3–60.7) | 51.9 (44.0–61.3) | 49.2 (38.5–63.0) | 54.7 (43.8–68.4) | 42.3 (34.6–51.7) | 47.7 (37.0–61.5) | 36.9 (26.9–50.7) |
| Ugunja | 1 | 132 (17.2%) | 46.2 (38.4–55.5) | 55.2 (44.5–68.5) | 36.9 (26.9–50.7) | 36.2 (28.8–45.4) | 50.0 (39.3–63.6) | 21.9 (13.8–34.8) | 24.2 (17.9–32.8) | 35.8 (26.0–49.4) | 12.3 (6.4–23.6) |
| Alego-Usonga | 1 | 146 (19.1%) | 58.9 (51.4–67.5) | 67.6 (57.6–79.4) | 50.7 (40.1–64.2) | 33.1 (25.8–42.4) | 34.8 (25.1–48.5) | 33.3 (22.9–48.6) | 27.9 (21.3–36.6) | 30.0 (21.0–42.9) | 27.1 (17.8–41.2) |
| Bondo | 1 | 113 (14.8%) | 37.2 (29.2–47.2) | 41.5 (31.1–55.4) | 31.9 (21.0–48.5) | 19.8 (13.6–28.8) | 25.0 (16.4–38.2) | 12.8 (6.0–27.0) | 18.8 (12.8–27.6) | 23.4 (15.1–36.5) | 12.8 (6.0–27.0) |
| Mobile clinic** | - | - | - | - | - | - | - | - | - | - | - |
| **Gender#** | | | | | | | | | | | |
| Male | - | 272 (50.5%) | - | 54.1 (43.4–67.4) | - | - | 38.5 (28.4–52.0) | - | - | 33.1 (24.6–44.5) | - |
| Female | - | 267 (49.5%) | - | 42.9 (31.3–58.7) | - | - | 30.5 (21.9–42.4) | - | - | 26.2 (18.1–37.9) | - |
| **Children age category#** | | | | | | | | | | | |
| <2 years | - | 295 (55.2%) | - | 40.3 (31.3–52.0) | - | - | 27.4 (20.5–36.7) | - | - | 22.5 (17.2–29.5) | - |
| 2–3 years | - | 165 (30.9%) | - | 64.8 (52.4–80.3) | - | - | 41.9 (33.3–52.6) | - | - | 38.8 (30.1–49.9) | - |

*(Continued)*

**Table 1.** (Continued)

| Factor | Total number of facilities | Total number of participants (%) | Positive rate by microscopy % (95%CI) | | | Positive rate by PCR % (95%CI) | | | Positive rate by both microscopy and PCR % (95%CI) | | |
|---|---|---|---|---|---|---|---|---|---|---|---|
| | | | Overall (n = 1026) | Children below five years (n = 542) | Pregnant women (n = 484) | Overall (n = 978) | Children below five years (n = 528) | Pregnant women (n = 450) | Overall (n = 1013) | Children below five years (n = 540) | Pregnant women (n = 473) |
| ≥4 years | - | 74 (13.9%) | - | 55.4 (38.5–79.7) | - | - | 44.4 (32.5–60.9) | - | - | 39.2 (27.0–56.9) | - |
| **Women age category$** | | | | | | | | | | | |
| 14–19 years | - | 97 (20.3%) | - | - | 37.1 (23.7–58.1) | - | - | 24.2 (14.2–41.1) | - | - | 18.5 (10.1–33.9) |
| 20–25 years | - | 181 (37.8%) | - | - | 33.1 (25.6–42.9) | - | - | 23.1 (16.7–31.9) | - | - | 18.0 (13.4–24.1) |
| 26–31 years | - | 119 (24.8%) | - | - | 24.4 (16.4–36.2) | - | - | 15.7 (7.4–33.4) | - | - | 10.3 (5.4–19.9) |
| 32–37 years | - | 58 (12.1%) | - | - | 22.4 (12.9–39.0) | - | - | 20.4 (8.1–51.2) | - | - | 10.3 (3.7–29.2) |
| 38–43 years | - | 24 (5.0%) | - | - | 37.5 (18.2–77.1) | - | - | 39.1 (17.5–87.4) | - | - | 29.2 (9.6–88.8) |
| **Educational level$** | | | | | | | | | | | |
| No education | - | 99 (21.3%) | - | - | 19.2 (11.5–32.1) | - | - | 13.2 (7.5–23.1) | - | - | 7.1 (3.2–16.1) |
| Primary | - | 262 (56.3%) | - | - | 30.5 (20.9–44.6) | - | - | 23.4 (12.2–44.6) | - | - | 17.1 (9.3–31.4) |
| Secondary | - | 89 (19.1%) | - | - | 43.8 (32.1–59.8) | - | - | 28.6 (17.7–46.2) | - | - | 23.5 (15.8–35.0) |
| Post-secondary | - | 15 (3.2%) | - | - | 26.7 (18.6–38.3) | - | - | 0 | - | - | 0 |
| **Marital status$** | | | | | | | | | | | |
| Single | - | 80 (17.2%) | - | - | 45.0 (35.8–56.5) | - | - | 27.3 (19.4–38.3) | - | - | 22.8 (16.9–30.7) |
| Married | - | 376 (80.9%) | - | - | 27.9 (19.3–40.4) | - | - | 20.8 (11.2–38.6) | - | - | 14.4 (8.0–26.0) |
| Separated/Widowed | - | 9 (1.9%) | - | - | 11.1 (1.4–86.4) | - | - | 0 | - | - | 0 |
| **Trimester$** | | | | | | | | | | | |
| 1st trimester | - | 40 (9.6%) | - | - | 37.5 (25.2–55.8) | - | - | 32.4 (18.3–57.5) | - | - | 23.7 (13.6–41.3) |
| 2nd trimester | - | 178 (42.7%) | - | - | 27.0 (20.2–35.9) | - | - | 21.8 (14.0–34.0) | - | - | 12.4 (7.4–20.9) |
| 3rd trimester | - | 199 (47.7%) | - | - | 29.1 (19.4–43.9) | - | - | 15.7 (8.5–28.9) | - | - | 14.1 (7.8–25.6) |

The age range for children below five years was <1 year to 4 years, and that for pregnant women was 14 to 43 years.

*Mobile clinics drew their participants from all across the particular county, since they were specialized clinics that traversed the entire county of operation.

**On the day of visit, the mobile clinic for Siaya County was not operational.

#These variables applied only to the data for children below five years.

$These variables applied only to the data for pregnant women.

-Indicates that data was not relevant/available for that variable.

their female counterparts 30.5% (95%CI: 21.9–42.4) (Table 1). Same pattern of infection was observed with microcopy results.

## Antenatal care attendance and IPTp-SP uptake by women during pregnancy

Of the 484 participating pregnant women, 384 (79.3%) reported to have previously started attending ANC prior to the survey day, with Kwale and Siaya counties showing a significant difference (p = 0.001); 101 (95.3%) and 184 (82.1%) respectively in the number of women who have attended at least one ANC visit. Significantly (p = 0.001) more number of pregnant women in Siaya County, 40 (17.9%), than Kwale, 5 (4.7%), had not started attending ANC, prior to the survey day. Additionally, many of the pregnant women surveyed made their first ANC visit in their second trimester, 144 (29.8%), or third trimester, 174 (36.0%) (Table 2).

Further, IPTp-SP treatment was reportedly administered during an ANC visit. Among the ANC attendees prior to the survey day, 325 (84.6%) reported to have started receiving IPTp-SP doses and 285 (68.4%) had reportedly received at least one dose of SP prior to the survey day, with significantly (p<0.001) higher proportion of pregnant women in Kwale (88.6%) than Siaya County (60.4%) receiving at least one dose. Additionally, among the pregnant women who had received at least one dose, only 61 (34.3%) reportedly took their first dose within the $2^{nd}$ trimester (14–26 weeks) as required while the remaining majority 81 (40.7%) received their first dose during the $3^{rd}$ trimester (27–40 weeks) (Table 2). Further, timing of the administration of the first IPTp-SP dose among the pregnant women showed that SP dose was first given on the second month, and majority of the women got their first dose between the sixth (18.7%) and eighth (25.3%) months. The findings demonstrated that the timing of first IPTp-SP dose was significantly correlated with the timing of the first ANC attendance (r = 0.33, p<0.001).

## Risk factors associated with malaria infection in children below five years and pregnant women

**Univariable analysis.** In overall, participants in Siaya County had significantly higher odds of malaria infection than those in Kwale County, OR = 2.40 (95%CI: 1.68–3.43), p<0.001. In Siaya County, significantly higher odds of malaria infection was observed in the

**Table 2. Comparison of the first antenatal care visit and intermittent preventive treatment (IPTp-SP) uptake by trimester of the pregnant women in Kwale and Siaya counties, Kenya.**

| Characteristics | Overall | Kwale County | Siaya County | Measuring differences among the characteristics by the counties$ [Diff (Z-test, p-value)] |
|---|---|---|---|---|
| *Trimester of $1^{st}$ ANC visit* | | | | |
| $1^{st}$ trimester | 21 (4.3%) | 2 (1.6%) | 19 (5.3%) | Diff = -0.04 (Z = -0.23, p = 0.819) |
| $2^{nd}$ trimester | 144 (29.8%) | 34 (26.6%) | 110 (30.9%) | Diff = -0.04 (Z = -0.48, p = 0.632) |
| $3^{rd}$ trimester | 174 (36.0%) | 66 (51.6%) | 108 (30.3%) | Diff = 0.21 (Z = 2.80, p = 0.005*) |
| *Trimester of $1^{st}$ IPTp-SP dose* | | | | |
| $1^{st}$ trimester | 3 (0.7%) | 1 (0.8%) | 2 (0.6%) | Diff = 0.00 (Z = 0.02, p = 0.987) |
| $2^{nd}$ trimester | 61 (12.6%) | 18 (14.1%) | 43 (12.1%) | Diff = 0.02 (Z = 0.21, p = 0.831) |
| $3^{rd}$ trimester | 81 (16.7%) | 31 (24.2%) | 50 (14.0%) | Diff = 0.10 (Z = 1.16, p = 0.245) |

$The statistical differences among the observed characteristics (i.e. first ANC visit and IPTp-SP uptake by trimester) by county was determined using test of proportions reporting the observed difference, Z-test statistic and p-value.

*Indicated a significant difference in the proportions, i.e. p<0.05.

following sub-counties when compared to Bondo: Ugenya (OR = 4.37 (95%C: 2.45–7.81), p<0.001), Alego-Usonga (OR = 2.00 (95%C: 1.10–3.63), p = 0.023), and Ugunja (OR = 2.29 (95%C: 1.27–4.12), p = 0.006). In Kwale County, significantly higher odds of malaria infection was observed in the following sub-counties when compared to Lunga Lunga: Matuga (OR = 7.25 (95%C: 1.99–26.35), p = 0.003) and Kinango (OR = 6.11 (95%C: 1.66–22.44), p = 0.006) (Table 3).

*Children below five years*. Univariable analysis for the factors associated with malaria infection in children below five years showed that children in Siaya County were marginally non-significantly at risk of malaria infection compared to those in Kwale County, OR = 1.51 (95% C: 0.98–2.31), p = 0.060. In Siaya County, we observed that children in Ugenya and Ugunja sub-counties were at significantly higher odds of malaria infection when compared to those in Bondo Sub-County. Further, we observed a gradient effect relative to the age of the child; children aged 2 to 3 years (OR = 1.91 (95%CI: 1.27–2.87), p = 0.002), and those aged 4 to below 5 years (OR = 2.12 (95%CI: 1.24–3.60), p = 0.006), were both significantly associated with higher odds of malaria infection compared to those aged below 2 years. However, no significant risk was observed by the gender of the children (Table 3).

*Pregnant women*. Univariable analysis for the factors associated with malaria infection in pregnant women showed that pregnant women in Siaya County were more than six times at risk of malaria infection compared to those in Kwale County, OR = 6.98 (95%C: 2.97–16.43), p<0.001. In Siaya County, significant risk was observed in Ugenya (OR = 8.25 (95%C: 3.07–22.15), p<0.001) and Alego-Usonga (OR = 3.42 (95%C: 1.22–9.54), p = 0.019) sub-counties when compared to Bondo Sub-County. Pregnant women with only primary (OR = 2.01 (95% C: 1.02–3.95), p = 0.043) or secondary (OR = 2.63 (95%C: 1.22–5.69), p = 0.014) level of education had significant risk of infection. Importantly, the results showed that pregnant women in their first trimester (OR = 2.58 (95%C: 1.17–5.68), p = 0.019), those who were presenting themselves for the first ANC visit (OR = 9.00 (95%C: 1.89–42.94), p = 0.006), as well as those who had not started taking IPTp-SP doses (OR = 4.44 (95%C: 1.64–12.02), p = 0.003) all had increased odds of malaria infection (Table 3).

## Multivariable analysis

Multivariable analysis of factors associated with malaria infection was conducted among pregnant women only as shown in Table 4. The analysis included factors such as the county of the participant, education level, trimester and ANC attendance. In overall, pregnant women in Siaya County had significantly higher odds of malaria infection, aOR = 6.30 (95%C: 2.40–16.51), p<0.001. In addition, pregnant women presenting themselves for their first ANC visit were significantly at risk, aOR = 5.40 (95%C: 0.96–30.50), p = 0.046.

## Discussion

The susceptibility of pregnant women and children below five years of age to malaria infection especially in the rural setting is well documented [26]. This study provides an up-to-date evaluation of malaria infection levels among pregnant women and children below five years of age attending lower levels of the health system (health centres and dispensaries) which typically manage the majority of childhood and pregnancy-related febrile disease burden in the Sub-Saharan Africa. The two rural counties sampled for this study are located within the moderate and high malaria endemic regions [4], where the burden of malaria parasitaemia is often very high among children below five years [7], and pregnant women [3]. Therefore, these findings are important to the country's NMCP and the county governments given that health is a

**Table 3.  Univariable analysis of factors associated with malaria infection (as measured using PCR technique) fitted separately for children below 5 years and pregnant women in Kwale and Siaya counties, Kenya.**

| Factor | Univariable analysis of malaria infection measured using PCR technique [OR (95%CI), p-value] | | |
|---|---|---|---|
| | Children below five years, (n = 528) | Pregnant women, (n = 450) | Overall, (n = 978) |
| **County** | | | |
| Kwale | Reference | | |
| Siaya | 1.51 (0.98–2.31), p = 0.060 | 6.98 (2.97–16.43), p<0.001* | 2.40 (1.68–3.43), p<0.001* |
| **Sub-County[&]** | | | |
| **Kwale** | | | |
| Matuga | 5.87 (0.67–51.20), p = 0.109 | 1.25 (0.16–9.67), p = 0.831 | 7.25 (1.99–26.35), p = 0.003* |
| Kinango | 7.88 (0.88–70.15), p = 0.064 | 0.43 (0.04–5.04), p = 0.502 | 6.11 (1.66–22.44), p = 0.006* |
| Msambweni | 3.27 (0.36–30.10), p = 0.295 | 0.46 (0.04–5.43), p = 0.540 | 3.24 (0.83–12.62), p = 0.091 |
| Lunga lunga | Reference | | |
| **Siaya** | | | |
| Rarieda | 1.00 (0.45–2.23), p = 1.000 | 1.52 (0.48–4.79), p = 0.476 | 1.14 (0.60–2.19), p = 0.689 |
| Gem | 1.53 (0.71–3.30), p = 0.272 | 1.64 (0.57–4.75), p = 0.362 | 1.46 (0.80–2.69), p = 0.221 |
| Ugenya | 2.91 (1.38–6.13), p = 0.005* | 8.25 (3.07–22.15), p<0.001* | 4.37 (2.45–7.81), p<0.001* |
| Ugunja | 3.00 (1.43–6.31), p = 0.004* | 1.91 (0.68–5.42), p = 0.222 | 2.29 (1.27–4.12), p = 0.006* |
| Alego-Usonga | 1.61 (0.75–3.43), p = 0.222 | 3.42 (1.22–9.54), p = 0.019* | 2.00 (1.10–3.63), p = 0.023* |
| Bondo | Reference | | |
| **Gender[#]** | | | |
| Male | 1.43 (0.99–2.05), p = 0.057 | - | - |
| Female | Reference | - | - |
| **Children age category[#]** | | | |
| <2 years | Reference | - | - |
| 2–3 years | 1.91 (1.27–2.86), p = 0.002* | - | - |
| ≥4 years | 2.12 (1.24–3.60), p = 0.006* | - | - |
| **Women age category[$]** | | | |
| 14–19 years | - | 1.25 (0.55–2.82), p = 0.598 | - |
| 20–25 years | - | 1.17 (0.55–2.49), p = 0.678 | - |
| 26–31 years | - | 0.73 (0.32–1.69), p = 0.464 | - |
| 32–37 years | - | 2.51 (0.86–7.31), p = 0.091 | - |
| 38–43 years | - | Reference | - |
| **Educational level[$]** | | | |
| No education | - | Reference | - |
| Primary | - | 2.01 (1.02–3.95), p = 0.043* | - |
| Secondary | - | 2.63 (1.22–5.69), p = 0.014* | - |
| Post-secondary | - | Insufficient obs | - |
| **Marital status[$]** | | | |
| Single | - | 1.43 (0.81–2.51), p = 0.216 | - |
| Married | - | Reference | - |
| Separated/Widowed | - | Insufficient obs | - |
| **Trimester[$]** | | | |
| 1st trimester | - | 2.58 (1.17–5.68), p = 0.019* | - |
| 2nd trimester | - | 1.50 (0.88–2.55), p = 0.134 | - |
| 3rd trimester | - | Reference | - |
| **ANC attendance[$]** | | | |
| First ANC visit | - | 9.00 (1.89–42.94), p = 0.006* | - |
| 2–3 ANC visits | - | 2.31 (0.53–10.15), p = 0.268 | - |

(*Continued*)

**Table 3.** (Continued)

| Factor | Univariable analysis of malaria infection measured using PCR technique [OR (95%CI), p-value] | | |
|---|---|---|---|
| | Children below five years, (n = 528) | Pregnant women, (n = 450) | Overall, (n = 978) |
| ≥ 4 ANC visits | - | Reference | - |
| IPTp-SP dosage$ | | | |
| No SP dose | - | 4.44 (1.64–12.02), p = 0.003* | - |
| < 3 SP doses | - | 1.89 (0.70–5.06), p = 0.207 | - |
| ≥ 3 SP doses | - | Reference | - |

The age range for children below five years was <1 year to 4 years, and that for pregnant women was 14 to 43 years.

*Indicates a statistically significant association, i.e. p<0.05.

&Mobile clinics were excluded here since only one was surveyed in Kwale County and that for Siaya County was not operational on the day of visit; additionally, these clinics draw their participants from all parts of the county.

#These variables applied only to the data for children below five years.

$These variables applied only to the data for pregnant women.

-Indicates that data was not relevant/available for that variable.

devolved function in Kenya. Further, they will contribute to improvement of policies on integration of malaria control approaches in rural health facilities.

According to our findings, the overall malaria positivity rate as measured by PCR was high (27.9%), and equally higher for both group of participants; 34.1% and 22.0% for children below five years and pregnant women respectively. The overall positivity rate was twice as high in Siaya (32.2%) than Kwale County (16.5%). The higher malaria positivity rate in children compared to pregnant women is not surprising as similar results were observed by van Eijk and colleagues [3] in their recent systematic review and meta-analysis, where they noted that malaria parasites can sequester in the placenta thus avoiding detection by diagnostic tests, and

**Table 4.** Multivariable analysis of factors associated with malaria infection (as measured using PCR technique) in pregnant women in Kwale and Siaya counties, Kenya.

| Factor | [aOR (95%CI), p-value] |
|---|---|
| County | |
| Kwale | Reference |
| Siaya | 6.30 (2.40–16.51), p<0.001* |
| Educational level | |
| No education | Reference |
| Primary | 1.50 (0.66–3.42), p = 0.332 |
| Secondary | 2.08 (0.85–5.11), p = 0.109 |
| Post-secondary | 0.35 (0.04–3.14), p = 0.349 |
| Trimester | |
| 1st trimester | 0.77 (0.23–2.53), p = 0.664 |
| 2nd trimester | 0.79 (0.43–1.48), p = 0.468 |
| 3rd trimester | Reference |
| ANC attendance | |
| First/current ANC visit | 5.40 (0.96–30.50), p = 0.046* |
| < 4 ANC visits prior to survey day | 1.90 (0.40–8.94), p = 0.417 |
| ≥ 4 ANC visits prior to survey day | Reference |

*Indicates a statistically significant association, i.e. p<0.05.

the detectable concomitant peripheral parasite prevalence can be lower than that in the placenta [27–29]. Further, the high positivity rate observed in Siaya County when compared to Kwale can be attributed to the differing ecological settings, rainfall patterns, temperature fluctuations, and malaria transmission intensities. In addition, Siaya County is located in the wider western Kenya region, which is a high malaria transmission area, receives high annual precipitation ranging between 1200mm and 1800mm [21].

The results pointed out at a heavy burden of malaria infection among the pregnant women especially those in their first trimester of pregnancy, those presenting themselves for ANC clinics for the first time and those who have not started receiving IPTp-SP dosage. Previous studies have shown that use of IPTp-SP during pregnancy greatly reduce malaria infection and its associated severe consequences, however, pregnant women are not protected throughout the entire pregnancy and are still likely to become infected between doses or after the final dose, more so, if other complementary protective measures like LLINs are not being used, or when malaria parasites become resistant to sulphadoxine-pyrimethamine [27,30,31]. The study also noted that malaria positivity rate was high among pregnant women aged, either 38 to 43 years or 14 to 19 years. Generally, pregnant women are highly susceptible to symptomatic malaria than their non-pregnant counterparts. The excess risk of malaria infection varies with age, gravidity, parity and immunity [3]. Primigravidae pregnant women are more prone to malaria and often have higher prevalence and parasitaemia density than other pregnant women of the same population [32]. Additionally, the size of the risk of infection increases with the age of the pregnant woman reflecting the cumulative exposure to malaria over time, with parity, and with pregnancy-specific immunity acquired after exposure to malaria in previous pregnancies [33].

The results on ANC attendance showed that only a small majority of the pregnant women have started attending ANC, prior to the survey day. We observed significantly higher attendance rate in Kwale than Siaya County. In line with the WHO recommendations, the Kenyan Ministry of Health recommends that pregnant women use LLINs, IPTp-SP, and receive prompt and effective diagnosis and treatment for malaria during ANC visits in order to prevent adverse consequences. The guideline recommends at least four ANC visits during the pregnancy, during which the pregnant woman would be administered with IPTp-SP dose at each visit. Even though this study did not fully explore complete ANC attendance and IPTp-SP dosage uptake due to limitations arising from our study design, previous studies [34] have shown that there is usually significantly low complete ANC attendance among pregnant women in the same study area.

Analysis of malaria risk factors showed that pregnant women in their first trimester of pregnancy and those presenting themselves for the first ANC visit or have not received IPTp-SP dosage, were all at significant risk of malaria infection. This is in line with previous studies, that have shown high malaria risk in pregnant women during first trimester and low risk in the third trimester [35,36]. Whilst, our study showed that age of the pregnant women was not significantly associated with malaria infection, previous studies have documented that age is an independent factor that appeared to be the principal influencer of malaria mostly in teenage pregnant women [35,37]. Similarly, other studies have reported significant positive association of malaria infection with complete ANC visits or uptake of IPTp-SP doses [38].

There is a general agreement that ANC interventions, skilled attendance during pregnancy and at birth, and management of complications after delivery are essential in tackling the high burden of maternal mortality in Sub-Saharan Africa [39]. In Kenya, the uptake rate of ANC services and complete ANC attendance has remained low despite a government policy on free maternity care [40–43], majority of pregnant women wait until they are in their second trimester to make their first ANC visit, while another substantial proportion present themselves only

in the third trimester [44,45]. Several barriers have been linked with this poor healthcare seeking behaviours among pregnant women especially in low income countries including Kenya. These barriers can be broadly categorized as low education level of the woman and her spouse, marital status, income, media exposure, and history of previous pregnancy-related complications as well as the underlying cultural and behavioural factors that underpin the apparent reticence towards healthcare seeking [46–48]. According to a study conducted in Siaya County [48], one of our study areas, key barriers to seeking maternal healthcare cited by participants included distance to the health facility coupled with poor road infrastructure and unreliable means of transport, and fees levied at the health facilities. Future designing of interventional health programmes should take into consideration these barriers that contribute to low healthcare seeking behaviours during pregnancy, child birth, and post-natal periods.

## Study limitations

This study was not without limitations. (1) Since the study targeted participants who were voluntarily coming to the health facility to seek treatment and care offered during the ANC services, our results may then be biased by the exclusion of pregnant women who were not able to come to the facility during the survey, or who were not yet enrolled for antenatal care, or delivered at home. However, according to local authorities this proportion is low due to good medical awareness, relatively reliable transport network to the health facilities, and the introduction of mobile clinic services by the Ministry of Health in 2014 [49]. (2) The fact that the study observed high malaria positivity rates among the participants could have been influenced by the data collection periods which happened within the high malaria transmission season in Kenya which is usually between April and July when long rains are usually experienced [21,22]. (3) The differences in the positivity rates between the two survey counties can be attributed to the different and varying infection transmission settings as well as the time gap of close to one month difference in data collection between the two counties. (4) Since we recruited pregnant women who were still carrying on with their scheduled ANC visits in a one-time descriptive cross-sectional study design, we therefore cannot conclusively and certainly determine their complete ANC visits as well as their complete IPTp-SP dosage uptake. (5) Lastly, even though microscopic examination of malaria remains the "gold standard" for laboratory diagnosis of malaria infection in malaria-endemic areas, this method has known limitations; it require a level of skill not available in many health facilities especially in remote rural areas where most malaria transmission occurs [50], lack of functional microscopes or electricity to run them [51], lack of standard reagents like stains [52], and high workload among the few available staff which may affect the quality of the results [51]. In addition, this technique is prone to species misidentification and the possibility of misdiagnosis which can be due to low parasitaemia [25]. In this study, we tried to overcome these microscopy limitations by; (i) ensuring that all the local laboratory technicians were well trained on the study procedures, (ii) performing double-reading of the slides by two independent microscopists, (iii) subjecting 10% of all the samples to re-examination to guarantee quality control, (iv) having an experienced laboratory technologist from KEMRI supervising the sample processing, and (v) conducting confirmatory test on each sample using PCR technique, which is a more sensitive and accurate method [25].

## Conclusion

The results suggest that despite evidence of ANC attendance and administration of IPTp-SP dosage during clinic visits, positivity rate of *P. falciparum* still remains relatively high among pregnant women and children below five years, an indication of the continual malaria

transmission in these two rural counties of Kenya. Based on these findings, multi-modal programmes targeting improvement of malaria control services within the rural health facilities are urgently needed in rural malaria endemic areas. There is urgent need to monitor and ensure delivery of quality malaria interventions offered to pregnant women in rural facilities. Further, based on these results, broad-based approaches rather than single-focused interventions, are needed to overcome the multiple challenges pregnant women face when seeking maternal care at the ANC facilities. The study also suggest a revamped education and awareness on the benefits of ANC attendance that go beyond routine treatment of medical conditions.

## Supporting information

**S1 Dataset.**
(XLSX)

**S1 File.**
(DOCX)

## Acknowledgments

We are grateful to the Division of Family Health, Ministry of Health, Nairobi, Kenya and the two County's Ministry of Health and their County Health Management Teams (CHMTs) for their unwavering support for this work. Additionally, we sincerely thank the management of the health facilities and the study participants for their support and cooperation during the study period. Special thanks goes to all members of the study team and field personnel for their commitment towards this work.

## Author Contributions

**Conceptualization:** Collins Okoyo, Stephen Munga, Sammy M. Njenga, Henry M. Kanyi.

**Data curation:** Collins Okoyo, Edward Githinji, Janet Masaku, Judy Mwai, Lilian Nyandieka, Henry M. Kanyi.

**Formal analysis:** Collins Okoyo, Henry M. Kanyi.

**Funding acquisition:** Sammy M. Njenga, Henry M. Kanyi.

**Investigation:** Edward Githinji, Ruth W. Muia, Sammy M. Njenga, Henry M. Kanyi.

**Methodology:** Collins Okoyo, Edward Githinji, Ruth W. Muia, Janet Masaku, Judy Mwai, Lilian Nyandieka, Stephen Munga, Sammy M. Njenga, Henry M. Kanyi.

**Project administration:** Ruth W. Muia.

**Supervision:** Collins Okoyo, Ruth W. Muia, Janet Masaku, Judy Mwai, Lilian Nyandieka, Stephen Munga, Sammy M. Njenga, Henry M. Kanyi.

**Validation:** Sammy M. Njenga.

**Writing – original draft:** Collins Okoyo.

**Writing – review & editing:** Collins Okoyo, Edward Githinji, Ruth W. Muia, Janet Masaku, Judy Mwai, Lilian Nyandieka, Stephen Munga, Sammy M. Njenga, Henry M. Kanyi.

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
