## [Decision Letter · Decision Letter 0]

30 Sep 2020

PONE-D-20-23758

Assessing malaria infection among pregnant women and children below five years of age attending rural health facilities of Kenya: A cross-sectional survey in two counties of Kenya

PLOS ONE

Dear Dr. Okoyo,

Thank you for submitting your manuscript to PLOS ONE. After careful consideration, we feel that it has merit but does not fully meet PLOS ONE’s publication criteria as it currently stands. Therefore, we invite you to submit a revised version of the manuscript that addresses the points raised during the review process.  Please be sure to address comments from Reviewer 2 as provided in a separate Word file.  If you are unable to access these, please do contact us.

We look forward to receiving your revised manuscript.

Kind regards,

R Matthew Chico, MPH, PhD

Academic Editor

PLOS ONE

Journal Requirements:

2. Thank you for including your ethics statement:  "The study protocol received ethical approval from KEMRI’s Scientific and Ethics Review Unit (SERU No. 3252). Additional approval was provided by the county-level health authorities after they were appropriately briefed about the study. At health facility level, individual consent for pregnant women and written parental consent for children below five years was obtained.".   

Please provide additional details regarding participant consent. In the ethics statement in the Methods and online submission information, please ensure that you have specified what type you obtained (for instance, written or verbal, and if verbal, how it was documented and witnessed). If your study included minors, state whether you obtained consent from parents or guardians. If the need for consent was waived by the ethics committee, please include this information.

4. Please include additional information regarding the survey or questionnaire used in the study and ensure that you have provided sufficient details that others could replicate the analyses. For instance, if you developed a questionnaire as part of this study and it is not under a copyright more restrictive than CC-BY, please include a copy, in both the original language and English, as Supporting Information.

5. Please confirm if your survey or questionnaire was pre-tested or validated. If these did not occur, please provide the rationale for not doing so.

Reviewers' comments:

Reviewer's Responses to Questions

**Comments to the Author**

1. Is the manuscript technically sound, and do the data support the conclusions?

Reviewer #1: Yes

Reviewer #2: Partly

2. Has the statistical analysis been performed appropriately and rigorously? 

Reviewer #1: No

Reviewer #2: No

3. Have the authors made all data underlying the findings in their manuscript fully available?

Reviewer #1: No

Reviewer #2: Yes

4. Is the manuscript presented in an intelligible fashion and written in standard English?

Reviewer #1: Yes

Reviewer #2: Yes

5. Review Comments to the Author

Reviewer #1: The manuscript by Okoyo et al entitled, “Assessing malaria infection among pregnant women and children below five years of age attending rural health facilities of Kenya: A cross-sectional survey in two counties of Kenya” describes the prevalence of Plasmodium falciparum infection in pregnant women and children under five years of age using results from cross-sectional surveys conducted at rural health facilities in two Kenyan counties. The primary finding was a PCR positivity rate for malaria of 34.1% in pregnant women and 22.0% for children under five, with positivity varying by geographic location. A majority of pregnant women (86.4%) reported attending ANC at least once, though 62.1% did not have their first visit until their third trimester, which explains why most women (75%) did not receive their first IPTp dose until their second or third trimester.

The authors thoroughly describe the methods employed for sample processing and examination and report extensively on the prevalence of Plasmodium falciparum infection in their study population. The analysis is limited in its assessment of risk factors and is primarily focused on descriptive statistics. This study contributes to the knowledge of malaria prevalence in these counties, providing a baseline against which to measure the effect of future malaria programs targeted towards pregnant women and children under five.

Major comments:

1. Further describe the knowledge gaps this manuscript aims to address. In the Introduction, there is substantial information about the status of malaria prevention efforts in Kenya, but limited explanation of what sets the study apart from other malaria surveys in the country. Additionally the abstract should state why the study was done.

2. The analysis of ANC utilization and SP-IPTp uptake should consider the trimester of the participants. For example in Line 246: Was the 8.2% of ANC attendees who completed ANC visits counted only from women late in the third trimester? If women are in their first or second trimester they would not be expected to have 4 ANC visits completed. Nor should women in the first trimester have received SP-IPTp. It is also not clear if women were recruited from the ANC clinic how could any of the participants could not be enrolled in antenatal care? This only seems possible if the study visit took place prior to the clinic visit and the visit that day was not included. One way to clarify this might be to stratify the current Table 2 by trimester of the woman enrolled. How to take this into account in the univariate and multivariable analyses should also be considered (see #3).

3. The multivariable analysis should be clarified and consider expanding. Specifically, were separate multivariable analyses conducted for children and pregnant women? If so, this should be clarified in the methods and the table. If not, then there should be separate analyses for these two group. Additional co-variates should be considered for the analysis of children. Was data collected on reprted net use? Are there trends in age? Were other socio-economic data collected? The description of which variabes were included in the multivariable analysis should also be clarified.

4. The manuscript is very well-written, but the text could be tightened up so that the detail of the findings are more easily accessible to the reader. The study design and results are very straightforward and do not require the length and level of detail in the text to convey the relevant messages. There substantial repetition of data in text and tables. For example, the paragraph in lines 210-8 could be removed as all of the data is presented in the table. Also, unless interventions are likely to be tailored at the sub-county level, the variability at the sub-county level could be acknowledged in the text, but the details and the analysis could be moved to a supplement.

Minor comments:

1. Abstract: Add that the study also assessed ANC utilization as stated in the last sentace of the main text Introduction.

2. Introduction: Consider updating the World Malaria Report references to the more recent reports.

3. Introduction: Check line 113, were only 8% of malaria cases in Kenya in under 5s? It seems likely that either the wording is unclear or the number is incorrect.

4. Methods: How were the women and children chosen? Were they pairs, meaning the pregnant women were there with an older child and so they were sampled together? If this were the case, primigravidae would be underrepresented or absent. Also because of natural birth spacing, younger children would be underrepresented.

5. Methods: For the sample size calculations, was there a target precision of estimates that was desired?

6. Methods: Please include the malaria transmission season of May and July in this area of Kenya.

7. Methods: Were the surveys based on a validated questionnaire, e.g. the Malaria Indicator Survey? If so, consider referencing the validated tool.

8. Methods: Were the microscopy slides read by two readers or just one?

9. Methods: Pf infection is defined as requiring both positive microscopy and PCR. However, much of the statistical analysis uses PCR positivity as the outcome. Consider clarifying the definitions of infection in the methods. Based on the data in presented in Table 1, there were many microscopy false positives or pcr false negatives (microscopy positivity > pcr positivity) and as expected some submicroscopic infection (pcr positivity > being both pcr and microscopy positive). Given that the results are primarily based on PCR, I assume the authors are more confident in their PCR data. If so, they should be more clear about their interpretation of their results and not leave the reader to make their own conclusions.

10. Methods: Please clarify the term “minimum generated variables” in line 194.

11. Results: Consider including a standard Table 1 showing the demographic characteristics of the study population overall and by county. In this table and in the text, months should be the unit of age for children.

12. Results: Could the month difference in the survey dates explain some of the differences in prevalence between the counties? If so, this might be acknowledged in the discussion.

13. Results and Tables: Please define all the variables used to calculate adjusted ORs

14. Table 2: Please add p-values to show statistically different distributions of variables by county. “Median gestational age in weeks (quartiles)” for each trimester is not useful and could be removed.

15. Results: The paragraph in lines 266-272 is methods, not results.

16. Discussion: Consider adding discussion of barriers to seeking care in these counties which need to be addressed based on this data.

17. Discussion: Consider adding details about the limitations of the microscopy results if in fact the authors have concerns about their utility.

18. Discussion/Conclusion: Consider speculating on what exactly should be done differently based on these results.

Editorial comments: This manuscript is well-written, a few small comments.

1. Abstract: In the Methods it is not clear if cluster random sampling was used to select the participants or the health centers. Please make the language clearer.

2. Abstract: Line 49, high”er” not highest

3. Abstract: Consider changing “transmission” in line 56 to “prevalence”

Reviewer #2: This study describes malaria status in pregnant women and young children in two counties in Kenya. The manuscript needs major revisions with attention in particular to points like inclusion and exclusion criteria of study participants, and sample size calculation and why a cluster random design was chosen. These aspects are weakly given or lacking in the manuscript.

6. PLOS authors have the option to publish the peer review history of their article (what does this mean?). If published, this will include your full peer review and any attached files.

Reviewer #1: No

Reviewer #2: **Yes: **Rukhsana Ahmed

---

## [Author Response · Author response to Decision Letter 0]

23 Mar 2021

We have addressed all the editor's and reviewers' comments and included the details of the revisions in a separate file named "Response to reviewers".

---

## [Decision Letter · Decision Letter 1]

26 Apr 2021

PONE-D-20-23758R1

Assessing malaria infection among pregnant women and children below five years of age attending rural health facilities of Kenya: A cross-sectional survey in two counties of Kenya

PLOS ONE

Dear Dr. Okoyo,

Thank you for submitting your manuscript to PLOS ONE. After careful consideration, we feel that it has merit but does not fully meet PLOS ONE’s publication criteria as it currently stands. Therefore, we invite you to submit a revised version of the manuscript that addresses the points raised during the review process.

We look forward to receiving your revised manuscript.

Kind regards,

R Matthew Chico, MPH, PhD

Academic Editor

PLOS ONE

Reviewers' comments:

Reviewer's Responses to Questions

**Comments to the Author**

1. If the authors have adequately addressed your comments raised in a previous round of review and you feel that this manuscript is now acceptable for publication, you may indicate that here to bypass the “Comments to the Author” section, enter your conflict of interest statement in the “Confidential to Editor” section, and submit your "Accept" recommendation.

Reviewer #1: (No Response)

Reviewer #2: All comments have been addressed

2. Is the manuscript technically sound, and do the data support the conclusions?

Reviewer #1: Partly

Reviewer #2: Yes

3. Has the statistical analysis been performed appropriately and rigorously? 

Reviewer #1: No

Reviewer #2: Yes

4. Have the authors made all data underlying the findings in their manuscript fully available?

Reviewer #1: Yes

Reviewer #2: Yes

5. Is the manuscript presented in an intelligible fashion and written in standard English?

Reviewer #1: Yes

Reviewer #2: Yes

6. Review Comments to the Author

Reviewer #1: The authors have not fully addressed several key components of my prior comments. Below in the reiteration of some key points that make the manuscript, in my opinion, unacceptable for publication as it is currently written.

Below are: my prior comments, >>Authors: denotes the authors reply, and **denotes my current responses and attempts to provide further clarification. Comments threads are separated by ____

_______

1. The analysis of ANC utilization and SP-IPTp uptake should consider the trimester of the

participants. For example in Line 246: Was the 8.2% of ANC attendees who completed ANC

visits counted only from women late in the third trimester?

>> Authors: This was counted among all study-enrolled women, but women who were found to have

completed their ANC visits were mostly in their third trimester.

**Women cannot have completed their ANC visits prior to the end of their third trimester. One cannot determine the completeness of their ANC care (number of ANC visits or number of IPTp doses) until delivery. This study design cannot answer this question unless you limit the analysis to women who are at full term. Women must either be followed throughout their pregnancies or must have retrospective data collection at the time of delivery.

a. If women are in their first or second trimester they would not be expected to have 4 ANC visits completed. Nor should women in the first trimester have received SP-IPTp.

>> Authors: It is true that women in first or second trimester would not be expected to have complete

ANC visits, but the idea here was to find out at what stage of pregnancy do women start ANC

visits and also start receiving IPTp-SP doses and compare this with the national guidelines.

**I agree you may answer the trimester or gestational age of ANC enrollment with this study design, but as above the completeness of ANC care cannot be assessed in this study unless you only analyze full term women.

b. It is also not clear if women were recruited from the ANC clinic how could any of the

participants could not be enrolled in antenatal care? This only seems possible if the study visit

took place prior to the clinic visit and the visit that day was not included. One way to clarify this

might be to stratify the current Table 2 by trimester of the woman enrolled. How to take this into

account in the univariate and multivariable analyses should also be considered (see #3).

>> Authors: Women were recruited immediately they arrived at the facility and prior to being attended to at the clinic, see lines 190-191. The ANC visit that day was not counted.

**Again, if women have come for antenatal care and you enroll them in your study you cannot count them as not having enrolled in antenatal care. If some of your participants did not follow through and complete the ANC visit they came to the clinic to receive, then that is a significant problem with the study. Based on the study design, you have systematically missed women who did not enroll in the ANC care.

**The results section on ANC attendance requires major revision and should be removed other than analyses of the timing of enrollment in antenatal care – with acknowledgement that this analysis is only among women who have presented for antenatal care and does not reflect the overall population.

________

9. Methods: Pf infection is defined as requiring both positive microscopy and PCR. However,

much of the statistical analysis uses PCR positivity as the outcome. Consider clarifying the

definitions of infection in the methods. Based on the data in presented in Table 1, there were

many microscopy false positives or pcr false negatives (microscopy positivity > pcr positivity)

and as expected some submicroscopic infection (pcr positivity > being both pcr and microscopy

positive). Given that the results are primarily based on PCR, I assume the authors are more

confident in their PCR data. If so, they should be more clear about their interpretation of their

results and not leave the reader to make their own conclusions.

>> Authors: Thanks for this comment. We had stated that Pf infection was first defined by microscopy then confirmed by PCR (see lines 251-253). As much as many of the malaria results in the text were referring to PCR results we had also stated and compared it with microscopy both in the text and tables. We have now put more clarification here, please see lines 253-254. Additionally, we have now outlined the limitations of microscopy technique that made us compared its results with PCR and subsequently relied more on PCR results, see lines 469-478.

**The fact that microscopy prevalence is higher than PCR still raises concern. Your definition of infection requiring PCR confirmation of microscopy makes sense as you state the in the methods. However, then the analyses should be done using that as the outcome, e.g. Tables 3 and 4 should have the outcome as confirmed infection (microscopy and PCR positive) as compared to PCR only. Or you should decide that you trust your PCR and that PCR is truly detecting some submicroscopic infection. That can be justified and used. The key is to state your definition, justify it, and then use it.

**The explanation in the Discussion of why routine health center microscopy may not be acurate is true. However, the methods section describes research grade microscopy – well trained technicians, double reading, support from a research facility. Thus, the description in the Discussion doesn’t fit the current microscopy results.

_______

4. Methods: How were the women and children chosen? Were they pairs, meaning the pregnant

women were there with an older child and so they were sampled together? If this were the case,

primigravidae would be underrepresented or absent. Also because of natural birth spacing,

younger children would be underrepresented.

>> Authors: The study did not pair pregnant women with children below five years. These were separate groups of participants who were coming to the health facility. But it was possible that some pregnant women were accompanied with their children who were below five years to the clinic. However, their data were not paired. We have now clarified this in the text, please see lines 188-

190.

**How the children were selected remains unclear. The methods state “children below five years of age attending ANC”. Were these children accompanying women to their ANC visit? Were they coming to immunization/other well care OR were they coming for sick visits? If they were coming for sick visits, then one might anticipate that the test positivity rate would be higher among a symptomatic population.

Reviewer #2: All comments from previous review has been adequately addressed. I have no further comments. The manuscript is acceptable for publication from my side.

7. PLOS authors have the option to publish the peer review history of their article (what does this mean?). If published, this will include your full peer review and any attached files.

Reviewer #1: No

Reviewer #2: **Yes: **Rukhsana Ahmed MD, PhD

---

## [Author Response · Author response to Decision Letter 1]

9 Jun 2021

We have addressed all the editor's and reviewers' comments and included the details of the revisions in a separate file named "Response to reviewers".

---

## [Decision Letter · Decision Letter 2]

24 Jun 2021

PONE-D-20-23758R2

Assessment of malaria infection among pregnant women and children below five years of age attending rural health facilities of Kenya: A cross-sectional survey in two counties of Kenya

PLOS ONE

Dear Dr. Okoyo,

Thank you for submitting your manuscript to PLOS ONE. After careful consideration, we feel that only part of your manuscript merits further consideration.  The ANC completion and IPTp coverage information presented appears to be invalid for reasons of study design and queried by peer-review.  These cannot be overcome.  However, if you would like to have your manuscript further considered by PLOS ONE, kindly remove the ANC completion and IPTp coverage information entirely - text and table content - and submit a revised version.    

We look forward to receiving your revised manuscript.

Kind regards,

R Matthew Chico, MPH, PhD

Academic Editor

PLOS ONE

Reviewers' comments:

Reviewer's Responses to Questions

**Comments to the Author**

1. If the authors have adequately addressed your comments raised in a previous round of review and you feel that this manuscript is now acceptable for publication, you may indicate that here to bypass the “Comments to the Author” section, enter your conflict of interest statement in the “Confidential to Editor” section, and submit your "Accept" recommendation.

Reviewer #1: (No Response)

2. Is the manuscript technically sound, and do the data support the conclusions?

Reviewer #1: Partly

3. Has the statistical analysis been performed appropriately and rigorously? 

Reviewer #1: No

4. Have the authors made all data underlying the findings in their manuscript fully available?

Reviewer #1: Yes

5. Is the manuscript presented in an intelligible fashion and written in standard English?

Reviewer #1: Yes

6. Review Comments to the Author

Reviewer #1: The authors have not addressed key components of my prior comments. Again these issues make the manuscript, in my opinion, unacceptable for publication as it is currently written. In this revision the authors have acknowledged the limitations of some of their analyses, but the misleading analyses remain in the abstract and the results sections.

My prior reviews have described the rationale for why some of the presented analyses are misleading. My recommendation is that these sections of the manuscript should be removed rather than discounted in the discussion. Specifically at least the following statements should be removed.

Abstract:

“Of all the pregnant women, only 27 (8.2%) had attained the recommended number of ANC attendance of at least four visits and only 55 (13.2%) had received the recommended number of intermittent preventive treatment with sulfadoxine-pyrimethamine (IPTp-SP) of at least three doses.”

Results:

“At the time of the study, only 27 women (8.2%) of the ANC attendees reported to have completed four ANC attendance, despite the fact that they were yet to reach the full-term of the pregnancy.”

“However, at the time of the survey, in overall, only a total of 55 (11.6%) of the pregnant women had received the recommended dosage of three or more doses of IPTp-SP (again noting that all these women were yet to deliver).”

Many of the same conclusions can be reached by focusing on results showing that many women enroll in ANC late and thus do not benefit from clearing infections early in pregnancy: “Additionally, many of the pregnant women surveyed made their first ANC visit in their second trimester, 119 (37.0%), and third trimester, 200 (62.1%).” And “majority of the women got their first dose (of SP) between the sixth (18.7%) and eighth (25.3%) months.”

My prior comments on other aspects of the manuscript that remain unaddressed are below, though the most critical component is above.

--------------

1. It is also not clear if women were recruited from the ANC clinic how could any of the

participants could not be enrolled in antenatal care? This only seems possible if the study visit

took place prior to the clinic visit and the visit that day was not included. One way to clarify this

might be to stratify the current Table 2 by trimester of the woman enrolled. How to take this into

account in the univariate and multivariable analyses should also be considered (see #3).

>> Authors: Women were recruited immediately they arrived at the facility and prior to being attended to at the clinic, see lines 190-191. The ANC visit that day was not counted.

**Revision 1: Again, if women have come for antenatal care and you enroll them in your study you cannot count them as not having enrolled in antenatal care. If some of your participants did not follow through and complete the ANC visit they came to the clinic to receive, then that is a significant problem with the study. Based on the study design, you have systematically missed women who did not enroll in the ANC care.

**Revision 1:The results section on ANC attendance requires major revision and should be removed other than analyses of the timing of enrollment in antenatal care – with acknowledgement that this analysis is only among women who have presented for antenatal care and does not reflect the overall population.

**Revision 2: “Pregnant women who had not started ANC visits were up to five times at risk of malaria infection, (adjusted odds ratio = 5.40, 95%CI: 0.96-30.50, p=0.046).” In fact these are women coming for their first ANC visit. These data do not include women who were in the community and not sought ANC care. Phrasing this as “Pregnant women presenting for their first ANC visit had up to five times higher odds of infection (aOR…) compared to women who had completed >= 4 ANC visits” is more accurate.

2. Methods: Pf infection is defined as requiring both positive microscopy and PCR. However,

much of the statistical analysis uses PCR positivity as the outcome. Consider clarifying the

definitions of infection in the methods. Based on the data in presented in Table 1, there were

many microscopy false positives or pcr false negatives (microscopy positivity > pcr positivity)

and as expected some submicroscopic infection (pcr positivity > being both pcr and microscopy

positive). Given that the results are primarily based on PCR, I assume the authors are more

confident in their PCR data. If so, they should be more clear about their interpretation of their

results and not leave the reader to make their own conclusions.

>> Authors: Thanks for this comment. We had stated that Pf infection was first defined by microscopy then confirmed by PCR (see lines 251-253). As much as many of the malaria results in the text were referring to PCR results we had also stated and compared it with microscopy both in the text and tables. We have now put more clarification here, please see lines 253-254. Additionally, we have now outlined the limitations of microscopy technique that made us compared its results with PCR and subsequently relied more on PCR results, see lines 469-478.

**Revision 1: The fact that microscopy prevalence is higher than PCR still raises concern. Your definition of infection requiring PCR confirmation of microscopy makes sense as you state the in the methods. However, then the analyses should be done using that as the outcome, e.g. Tables 3 and 4 should have the outcome as confirmed infection (microscopy and PCR positive) as compared to PCR only. Or you should decide that you trust your PCR and that PCR is truly detecting some submicroscopic infection. That can be justified and used. The key is to state your definition, justify it, and then use it.

**Revision 1: The explanation in the Discussion of why routine health center microscopy may not be acurate is true. However, the methods section describes research grade microscopy – well trained technicians, double reading, support from a research facility. Thus, the description in the Discussion doesn’t fit the current microscopy results.

3. Methods: How were the women and children chosen? Were they pairs, meaning the pregnant

women were there with an older child and so they were sampled together? If this were the case,

primigravidae would be underrepresented or absent. Also because of natural birth spacing,

younger children would be underrepresented.

>> Authors: The study did not pair pregnant women with children below five years. These were separate groups of participants who were coming to the health facility. But it was possible that some pregnant women were accompanied with their children who were below five years to the clinic. However, their data were not paired. We have now clarified this in the text, please see lines 188-

190.

**Revision 1:How the children were selected remains unclear. The methods state “children below five years of age attending ANC”. Were these children accompanying women to their ANC visit? Were they coming to immunization/other well care OR were they coming for sick visits? If they were coming for sick visits, then one might anticipate that the test positivity rate would be higher among a symptomatic population.

7. PLOS authors have the option to publish the peer review history of their article (what does this mean?). If published, this will include your full peer review and any attached files.

Reviewer #1: No

---

## [Author Response · Author response to Decision Letter 2]

19 Jul 2021

We have addressed all the editor's and reviewers' comments and included the details of the revisions in a separate file named "Response to reviewers".

---

## [Decision Letter · Decision Letter 3]

25 Aug 2021

PONE-D-20-23758R3

Assessment of malaria infection among pregnant women and children below five years of age attending rural health facilities of Kenya: A cross-sectional survey in two counties of Kenya

PLOS ONE

Dear Dr. Okoyo,

Thank you for submitting your manuscript to PLOS ONE. After careful consideration, we feel that it has merit but does not fully meet PLOS ONE’s publication criteria as it currently stands; Reviewer 1 has some valuable suggestions that will strengthen the manuscript.. Therefore, we invite you to submit a revised version of the manuscript that addresses the points raised by Reviewer 1.

We look forward to receiving your revised manuscript.

Kind regards,

R Matthew Chico, MPH, PhD

Academic Editor

PLOS ONE

Journal Requirements:

Reviewers' comments:

Reviewer's Responses to Questions

**Comments to the Author**

1. If the authors have adequately addressed your comments raised in a previous round of review and you feel that this manuscript is now acceptable for publication, you may indicate that here to bypass the “Comments to the Author” section, enter your conflict of interest statement in the “Confidential to Editor” section, and submit your "Accept" recommendation.

Reviewer #1: All comments have been addressed

2. Is the manuscript technically sound, and do the data support the conclusions?

Reviewer #1: Yes

3. Has the statistical analysis been performed appropriately and rigorously? 

Reviewer #1: Yes

4. Have the authors made all data underlying the findings in their manuscript fully available?

Reviewer #1: Yes

5. Is the manuscript presented in an intelligible fashion and written in standard English?

Reviewer #1: Yes

6. Review Comments to the Author

Reviewer #1: The authors have made many improvements in this last revision. The suggestions below are minor, but will still strengthen the manuscript prior to publication.

1.Rephrase line 295 to reflect, women presenting for their first ANC visit – not women who had not started ANC care.

2. Include the factors in line 353 and 354 in Table 3.

3. Tables 2 and 4 should include the number of participants in each category. In Table 2 the “n” can be in the column headers, but for Table 4 the ideal would be to but the break down by each “Factor”, for example Kwale (n=XXX), Siaya (n=YYY). This is especially helpful to see the break down for ANC attendance.

4. Remove “and have not started attending ANC clinics” in line 386.

7. PLOS authors have the option to publish the peer review history of their article (what does this mean?). If published, this will include your full peer review and any attached files.

Reviewer #1: No

---

## [Author Response · Author response to Decision Letter 3]

26 Aug 2021

Comments to the Author

1. If the authors have adequately addressed your comments raised in a previous round of review and you feel that this manuscript is now acceptable for publication, you may indicate that here to bypass the “Comments to the Author” section, enter your conflict of interest statement in the “Confidential to Editor” section, and submit your "Accept" recommendation.

Reviewer #1: All comments have been addressed

2. Is the manuscript technically sound, and do the data support the conclusions?

Reviewer #1: Yes

3. Has the statistical analysis been performed appropriately and rigorously? 

Reviewer #1: Yes

4. Have the authors made all data underlying the findings in their manuscript fully available?

Reviewer #1: Yes

5. Is the manuscript presented in an intelligible fashion and written in standard English?

Reviewer #1: Yes

6. Review Comments to the Author

Reviewer #1: The authors have made many improvements in this last revision. The suggestions below are minor, but will still strengthen the manuscript prior to publication.

1.Rephrase line 295 to reflect, women presenting for their first ANC visit – not women who had not started ANC care.

>> Thanks for pointing this out, we have made the change.

2. Include the factors in line 353 and 354 in Table 3.

>> In the last review, we were advised to remove these components of ANC in the table. Nonetheless, we have included this additional suggestion.

3. Tables 2 and 4 should include the number of participants in each category. In Table 2 the “n” can be in the column headers, but for Table 4 the ideal would be to but the break down by each “Factor”, for example Kwale (n=XXX), Siaya (n=YYY). This is especially helpful to see the break down for ANC attendance.

>> The number of participants are already summarized in table 1, are you suggesting that we repeat them in these tables?

>> Actually, in table 2, what we presented are raw numbers with percentages in bracket, so we don’t think there are other numbers to present unless we don’t clearly get your suggestion.

4. Remove “and have not started attending ANC clinics” in line 386.

>> We have removed this statement.

---

## [Editor Report · Decision Letter 4]

31 Aug 2021

Assessment of malaria infection among pregnant women and children below five years of age attending rural health facilities of Kenya: A cross-sectional survey in two counties of Kenya

PONE-D-20-23758R4

Dear Dr. Okoyo,

We’re pleased to inform you that your manuscript has been judged scientifically suitable for publication and will be formally accepted for publication once it meets all outstanding technical requirements.

Kind regards,

R Matthew Chico, MPH, PhD

Academic Editor

PLOS ONE
---

## [Editor Report · Acceptance letter]

8 Sep 2021

PONE-D-20-23758R4 

Assessment of malaria infection among pregnant women and children below five years of age attending rural health facilities of Kenya: A cross-sectional survey in two counties of Kenya 

Dear Dr. Okoyo:

I'm pleased to inform you that your manuscript has been deemed suitable for publication in PLOS ONE. Congratulations! Your manuscript is now with our production department. 

Kind regards, 

on behalf of

Dr. R Matthew Chico 

Academic Editor

PLOS ONE